# Learning Experiences of Future Healthcare Support Workers Enrolled in a Dual Mode Vocational Training Programme

**DOI:** 10.3390/healthcare11192678

**Published:** 2023-10-03

**Authors:** María Dolores Ruiz-Fernández, Iván Gámez-Vargas, María Isabel Ventura-Miranda, Iria Dobarrio-Sanz, María del Mar Jiménez-Lasserrotte, Ángela María Ortega-Galán

**Affiliations:** 1Department of Nursing, Physiotherapy and Medicine, University of Almeria, 04120 Almeria, Spain; mrf757@ual.es (M.D.R.-F.); ids135@ual.es (I.D.-S.); mjl095@ual.es (M.d.M.J.-L.); 2Facultad de Ciencias de la Salud, Universidad Autónoma de Chile, Temuco 1090, Chile; 3IES Albaida, 04009 Almeria, Spain; ivangv333@gmail.com; 4Department of Nursing, University of Huelva, 21071 Huelva, Spain; angelaortega96@gmail.com

**Keywords:** qualitative research, experiences, healthcare support workers, learning, professional training

## Abstract

Background: With life expectancy increasing, there is a growing need to train healthcare support workers who provide care for dependent people in healthcare centres and at home. This qualitative study, based on Gadamer’s hermeneutic philosophy, aimed to understand the learning experiences of future healthcare support workers currently enrolled in an intermediate, dual modality vocational training programme with regard to caring for dependent people. Methods: Convenience sampling was used to recruit the participants, who were all students enrolled in an intermediate level vocational training programme in care for dependent people. Fourteen in-depth interviews and one focus group session were conducted with the students. Atlas.ti 8.0 software was used to analyse the participants’ accounts. Results: The students highlighted the vocational nature of their studies and the need to feel competent and useful as a healthcare support worker for dependent people. Practice-based learning and the need for training in core competences are complementary and essential elements of the training process. Conclusions:The participants’ previous experiences were key in determining their academic trajectory and reflect their motivation and interest to learn. However, they feel vulnerable, unprotected, and lack training in psychosocial skills. Educational institutions should focus training programmes on the practice and development of psychosocial skills that motivate students to acquire transversal competences.

## 1. Introduction

Rising life expectancy is increasing the need to provide care for vulnerable and dependent older adults [1]. By 2050, 38% of the world’s population is expected to be over 65 years of age [2]. Spain follows the global trend and is one of the fastest ageing countries along with Portugal, Greece, Slovenia, Korea, and Poland [3]. A total of 19.77% of the Spanish population is over 65 years of age [4], of which 30% are dependent [5] from the Spanish population, 19.77% are over 65 years old.

Although family caregivers provide most of the care needed by these individuals [6], this situation results in a rapidly growing need for domestic care services and social and healthcare facilities [7]. Healthcare support workers are essential in optimising the quality and safety of care provided at home and in care homes [8]. They are responsible for attending to the physical and emotional needs of patients in different social and healthcare settings [9]. In Spain, the Royal Decree 1593/2011 of 4th November 2011 recognises the figure of Technician in Care for People in a Situation of Dependency (TCPSD), otherwise known as a healthcare support worker, within the area of vocational training (VT). This is an intermediate level training programme lasting 2000 h, in which the trained professional will be able to work in the service sector, providing care, psychosocial support, and assistance in household management [10]. The gradual increase in dependent people in Spain and the consequent need for care [11] explains why there are currently 19,856 students enrolled in this training programme [12]. Furthermore, since 2012 there has been the option of offering VT qualifications in the dual mode, in which companies play a leading role in the training process [13]. This educational methodology is based on students spending time in companies in the sector, which offers them practice-based learning in a real workplace [14]. Therefore, the aim of this study was to understand the learning experiences of future healthcare support workers currently enrolled in an intermediate, dual modality vocational training programme with regard to caring for dependent people. The professional profile of the degree is determined by its general, professional, personal, and social competences, and by the list of qualifications of the national catalogue of professional qualifications included in the degree. The professional subjects are the organization of care for people in situations of dependency, social skills, characteristics and needs of people in situations of dependency, psychosocial care and support, communication support, home support, healthcare, hygienic care, telecare, first aid, training and career guidance, and company and entrepreneurship training in workplaces.

## 2. Materials and Methods

### 2.1. Design

A qualitative study was designed based on Gadamer’s hermeneutic philosophy (Hans-Georg Gadamer 1900–2002 [15]), which highlights the dialogical character of understanding. Language allows us to reveal the human essence: through linguistic interpretation we gain access to truth and knowledge of the world [16,17,18]. For Gadamer, it is not possible to disregard the researcher’s prior conceptions. Therefore, researchers must ensure that the phenomenon they intend to study can be experienced in the lifeworld and they must clarify their prior understanding of the phenomenon [16,19,20]. The study followed the consolidated criteria for qualitative research reporting (COREQ). The researchers were a man and a woman, both teachers with many years of experience. The relationship with the participants: they were students included in vocational training programmes in dual mode where the researchers taught. The method of approaching the participants: convenience sampling, due to the availability of the sample [21]. Phenomenology is a form of qualitative research that focuses on the study of an individual’s lived experiences in the world. For Gadamer, human experience cannot be understood without language. Following the Gadamerian method developed by Valerie Fleming, it was possible to understand the stories of the participants through a dialogue, from which meanings emerged. Understanding the stories of the participants requires being prepared to be told something through a dialogue, from which meanings emerge [15].

### 2.2. Participants

The participants were students of the intermediate level training programme TCPSD in a secondary school (SS) in southeastern Spain. The researchers used convenience sampling methods to recruit the students, given the accessibility of the sample. The participants had to meet the following inclusion criteria: (1) Be enrolled and active students in any of the two courses of the TCPSD training programme; (2) Have given their consent to participate in the study. The study sample consisted of 23 students enrolled in the 1st and 2nd years of TCPSD, with an average age of 20 years (SD = 6.34) and an average professional experience in caring for dependent people in other settings (home help service, family care, etc.) of 19.6 months (SD = 36) (Table 1).

### 2.3. Data Collection

Data were collected during the months of January and February 2022. For this purpose, 9 students participated in a focus group; 14 in-depth interviews were conducted. It approached 26 students, but only 23 students gave consent. The study was conducted in a classroom within the secondary school (SS) where the participants were studying. The focus group was conducted by two researchers as a first step in the exploratory phase of the study. One researcher acted as the moderator and the other as the observer. The focus group lasted approximately 60 min. The interviews were conducted by one researcher and lasted approximately 45 min. Both approaches involved taking notes with pre-analytical intuitions. Both the focus group and the interviews were audio-recorded and transcribed with the consent of the participants. Data collection was completed when data saturation was reached. The focus group and interviews followed an interview script developed by the research team (Table 2).

### 2.4. Analysis

The transcripts and field notes were incorporated into a hermeneutic unit and analysed by two researchers with experience in teaching VT students. The analysis followed the steps described by Fleming [17]. Firstly, the researchers established a dialogue with the participants by taking notes that were useful in the transcription and subsequent coding process. Secondly, the researchers gained an understanding of the phenomenon studied through analysing the transcripts. This entailed reading the transcripts to get a general idea of what the participants said. Subsequently, the transcripts were reread line by line to carry out a more detailed analysis. Lastly, significant extracts were selected as quotations and assigned codes to reflect their meaning. The codes were then grouped into units of meaning, subthemes, and themes. Version 8 of the ATLAS.ti software was used to support the analysis.

### 2.5. Rigor

Rigour was ensured by following Lincoln and Guba’s criteria [22]:(1)Credibility: The data collection process was described in detail and a researcher supported the interpretation of the data. All of the participants’ opinions and views were collected and they verified the units of meaning, themes, and subthemes. Lastly, the analytical process was checked by two independent reviewers;(2)Transferability: the method, participants, setting, and context of the study were described in detail;(3)Reliability: three researchers, who were not involved in data collection and had experience in qualitative research and student training, corroborated the analysis;(4)Confirmability: the analysis was subsequently validated by the study’s future healthcare support workers, who confirmed the accuracy and interpretation of the transcripts.

## 3. Results

Three main themes and six subthemes were extracted that helped describe and understand the learning experiences of students enrolled in the dual vocational training for future dependent health support workers. (Table 3).

### 3.1. The Role of the Healthcare Support Worker in Caring for Dependent People

In the context of dependency, the healthcare support worker is essential in providing care and support to people in need.

#### 3.1.1. Autonomy as a Synonym for Success

The participants acknowledge that their main professional responsibility is related to promoting dependent people’s autonomy, as well as maintaining and improving their quality of life on a day-to-day basis.


*“(…) may both support workers and dependants be treated well and have as much personal autonomy as possible and not be harmed in any way.”*
I14

The participants stated that their role goes beyond the healthcare they already provide and should also focus on domestic management and psychosocial support tasks.


*“It’s about improving the patient’s quality of life and condition by providing both physical and psychological help with all means possible. Not only do we provide the person care, but we also treat them psychologically.”*
FG6


*“(...)I play a role in cleaning the dependent person’s home and maintaining their personal hygiene.”*
I4

#### 3.1.2. The Positive Social Impact of Educational and Professional Success as a Result of a Vocational Calling

A large majority of the statements provided emphasised the importance of vocation as the key to educational and professional success. According to the participants, this professional calling is defined by a set of values that are essential in a healthcare support worker and which will have an impact on providing more humanised care.


*“(...) this course is based on empathy, assertiveness and putting yourself in someone else’s shoes. If you like working with people who are disadvantaged or have some kind of dependency, I recommend this course as this is the foundation of your daily professional life.”*
FG4


*“It will only be difficult for you if you don’t have a vocation, because in this job, from my point of view, it’s all about vocation.”*
I9

In this regard, the participants stated that they can become different types of healthcare support workers based on their own professional and personal experiences. Caring for this kind of users enriches and strengthens them on a professional, vocational, and even personal level. They find it so gratifying to provide care to these individuals with special needs that the feelings of satisfaction between the carer and the dependent person become mutual. This ultimately has positive repercussions on the healthcare support worker’s learning process.


*“Users with special educational needs are complicated but it is very rewarding to work with them as they show you that they love you or how you have taught them things that they can now do well.”*
FG9


*“(...) thanks to my training, I have made positive changes to the way I care for people with special educational needs within my own family. I have learned new aspects of care that improve their day to day life.”*
FG3

### 3.2. The Training Process of the Healthcare Support Worker

The participants stated that in order to carry out the role of a healthcare support worker successfully, they must undergo the vitally important stage of training. To do so, they must acquire a foundation of knowledge, skills, and attitudes that will enable them to deal with caring for a dependent person from a professional and multidisciplinary perspective.

#### 3.2.1. Practice-Based Learning

In their opinion, this setting gives them the opportunity to gain hands-on experience and an in-depth understanding of the reality of their role. They stressed the positive value of this experience as it guides them through their whole educational journey.


*“It would be a good idea to allocate more places on the dual programme so that more students can have first-hand professional experience.”*
I10


*“(...) if I had not studied on the DUAL programme, I would never have seen the reality of this course and what it is really about. You don’t really know what it’s like until you get to the internship. It changes your perspective and changes the way you think.”*
FG8

They stated that learning content from a practical point of view enables them to advance in the curriculum, since they are more adapted to the reality of their future role. This factor even leads them to undervalue the theoretical content.


*“(...) by experiencing the practical reality of this profession you end up empathising more, which makes it more enjoyable, practical and easier to do this course.”*
FG4


*“You can regurgitate theory in an exam, but if you don’t know how to do it in practice, the theory is of little use.”*
FG8

#### 3.2.2. Training as the Foundation of Professional Success

The participants referred to training as the foundation for them to be able to perform their role as a healthcare support worker. They feel motivated to do so and described that, despite investing a lot of time in training, they find it rewarding as it will have an impact on their professional career.


*“We need to be quite well-prepared and trained, as we spend many hours doing something we enjoy, which has a positive influence on our daily professional life.”*
FG4


*“We need to be fully prepared. That’s what we train for. We must undergo a teaching process that prepares us for all types of patients and teaches us to work in a team. This will produce very good professionals.”*
FG6

According to the participants, there are inevitably aspects to be improved in the educational process. With regard to the training programme, the participants put forward proposals for improvement in terms of its curricular development. These enable them to acquire certain core competences that are necessary to care for dependent people or can even be extrapolated to everyday situations.


*“For example, first aid could be more in-depth. I have a little brother and, before doing the course, I was afraid of being alone with him in case he drowned or whatever, and thanks to first aid I have learnt how to do CPR and the Heimilich manoeuvre.”*
FG3


*“It would be interesting to allocate more teaching hours to more important competences such as personal hygiene, which is what we work on daily with the user.”*
FG6

### 3.3. Vulnerability of the Healthcare Support Worker: Risks and Needs

The participants defined themselves as a vulnerable group exposed to a series of risks and situations associated with caring for dependent people. In addition to physical fatigue, the participants mentioned physical assaults and the emotional burden as two factors of professional burnout intrinsic to their future profession. They spoke of their own limits and fears. They also expressed the need for psychological support and training to help them avoid interference between their professional and personal lives.

#### 3.3.1. The Risk of Assault: An Unsafe Reality

The group openly referred to the fear of physical assault among the main risk factors associated with their work and which they experience during training.


*“I am afraid of them hitting me and doing something serious to me. Despite this, I deal with it on a daily basis and try to avoid it affecting my work.”*
FG3


*“I experienced assault in the care home I was in: bites, scratches, punches at the hands of the residents who were psychologically unwell.”*
FG6

The participants recognised that providing direct care and being so close to the dependent individuals has certain negative consequences; they can suffer or endure all of the dependents’ frustrations. This situation is a source of fear and uncertainty for them.


*“One of our biggest fears is that we are an easy target for the frustrations they have and we are the ones they are dealing with.”*
FG6

In this regard, according to the participants, everyone must recognise their limits in order to avoid further occupational risks. They made reference to feeling undervalued and socially unprotected.


*“The individual’s abilities and physical strength have an influence. For example, I sometimes find it difficult to carry out my professional duties on the patient because I am not as strong as I would like.”*
FG4


*“It is a very important profession and very undervalued socially.”*
I2

#### 3.3.2. The Risk of Emotional Overload and the Need to Learn to Manage Emotions

The participants asked for training in managing emotions and communication skills in order to be able to prevent the emotional overload associated with caring for patients in the medium and long term. They requested counselling and training that could, in their view, prevent burnout, especially for healthcare support workers for whom the psychological impact of their professional duties gradually undermines their professional work.


*“Healthcare support workers should be trained or taught to be psychologically strong. They need to be given more techniques and skills to develop empathy and be able to provide help without getting too involved, separating the professional from the personal, always focusing on helping the patient. You can’t just listen to the wall, saying “yes yes I am listening to you” while you look away (...)“.*
FG8


*“(...) reinforcing my skills in managing my emotions would help me because it would be easier for me to deal with patients and I would avoid being so affected by their problems on a psychological level”*
I10

## 4. Discussion

The aim of this study was to understand the learning experiences of future healthcare support workers currently enrolled in an intermediate, dual modality vocational training programme with regard to caring for dependent people. The participants highlighted the need to feel useful and competent in their role as a healthcare support worker.

A healthcare support worker provides support to people in need as reflected in other studies, such as Wu et al. [23]. Their tasks include a range of professional, personal, and social competencies that demonstrate how healthcare support workers help with health management activities, care organisation, domestic support and management, psychological support, medication administration, and increased demand for healthcare [24]. Human beings are living for longer than ever before, which leads to an increased demand for healthcare [25]. It is, therefore, necessary to professionalise the care sector in order to offer quality health services [26].

Practice-based learning and the dual training mode in core competences are complementary and indispensable elements of the training process. Our participants consider themselves competent professionals thanks to this training programme. They advocate the need for vocational training as an indispensable prerequisite to providing quality care adapted to each patient’s individual needs. This context allows for significant learning in caring for dependent people [27]. In fact, results have shown that dual vocational training leads to higher rates of employability than classroom-based vocational training [13]. This may be due to the fact that acquiring knowledge about caregiving through hands-on experience in the workplace is much more interesting and motivating.

According to our participants, this whole formative process is influenced by their previous experiences. They unanimously consider their experiences as family caregivers to be decisive in their academic trajectory, as reflected in other studies where the family plays an important role in caring for the patient [28]. Participants referred to how the need to provide direct care to a close family member drove them to make the decision to train in this field. Through comparison with other studies, we know that family caregivers often manage complex medical and nursing tasks [29] and often feel unprepared to cope with them [30]. In this respect, the participants were glad to have acquired a knowledge base that enables them to deal with caring for a patient from a professional and multidisciplinary perspective.

Providing care affects the healthcare support worker’s quality of life in physiological, psychological, financial, and social terms, as confirmed by other studies [31]. In light of this, it is clearly necessary to professionalise this sector. In line with our research, other studies have also shown how providing accurate information and knowledge can help alleviate the increasing physical and psychological burden of informal care [32].

As this research shows, the participants themselves define themselves as a helpless and vulnerable group and feel unsafe; they feel defenceless, unprotected, and undervalued on a social level. This is nothing new, as recent studies show that we should be more concerned about healthcare support workers [33], who need to develop personalised care practices based on patients’ individual preferences and needs [34]. This requirement leads the healthcare support worker to change their lifestyle and face different stressors that they are sometimes unable to control, which triggers a state of physical and mental exhaustion, thus hindering their performance in the work environment [9]. Some participants reported feeling psychologically affected due to the great emotional burden of caring for a patient [35]. The psychological impact of this profession on their own lives makes the participants uncomfortable.

Limitations and Future Research

The majority of participants are women, which could have influenced the study results. However, the reality is that the majority of health support workers are women, which is characteristic of this profession. Additionally, most participants had previous experience as informal caregivers for immediate family members.

## 5. Conclusions

The results of this study demonstrate the importance of specialized training for healthcare support workers. Individual factors, such as vocation, are key in the role of healthcare support workers who provide care to dependent people. Practice-based learning and the need to acquire core competences are complementary and essential to the training process. The participants’ experiences sparked their interest, motivated them, and were key in determining their academic trajectory. Nonetheless, they feel vulnerable, unprotected, and lacking in emotional management skills when caring for vulnerable groups such as older adults or people with disabilities.

With regard to future research, there is a need to delve deeper into the training needs of healthcare support workers as well as the students’ quality of life. This would allow us to develop tools or intervention programmes adapted to training requirements that would promote the acquisition of psychosocial skills and abilities in managing emotions.

## Figures and Tables

**Table 1 healthcare-11-02678-t001:** Sociodemographic data of the participants.

Participant	Age (Years)	Sex	Marital Status	Prior Experience (Months)	VT Course
P1-FG1	18	Male	Single	0	2
P2-FG2	19	Male	Single	50	2
P3-FG3	18	Female	Single	72	2
P4-FG4	18	Male	Single	8	2
P5-FG5	19	Male	Single	10	2
P6-FG6	30	Male	Divorced	156	2
P7-FG7	20	Female	Single	0	2
P8-FG8	18	Female	Single	0	2
P9-FG9	20	Female	Single	10	2
P10-I10	38	Female	Divorced	0	1
P11-I11	38	Female	Married	12	1
P12-I12	18	Female	Single	36	1
P13-I13	18	Female	Single	0	1
P14-I14	16	Female	Single	0	1
P15-I15	18	Female	Single	36	1
P16-I16	16	Female	Single	0	1
P17-I17	17	Female	Single	2	1
P18-I18	16	Female	Single	48	1
P19-I19	16	Female	Single	2	1
P20-I20	19	Female	Single	3	1
P21-I21	17	Female	Single	3	1
P22-I22	17	Female	Single	0	1
P23-I23	16	Female	Single	3	1

Note: FG = focus group; I = in-depth interview; VT = vocational training.

**Table 2 healthcare-11-02678-t002:** Interview script and guide.

Context	I am a researcher on this study. Our aim is to gain insight into the learning experiences of future healthcare support workers currently enrolled in intermediate-level vocational training, in relation to caring for dependent people.
Introduction and ethical considerations	Participation is voluntary. You are free to withdraw from the study at any time. Interviews will be recorded. We guarantee anonymity and data confidentiality. Verbal acceptance and signature of informed consent.
Starting question	Tell me about the experiences or situations that led you to study on this vocational training programme.
Development	What have your experiences been as a support worker for dependent people? How do your experiences influence how you care for dependent people? Do you think your experiences influence your involvement in the teaching–learning process? What do you think the main obstacles are in providing care to dependent people? How have you tried to overcome them? How do your experiences influence your decision-making skills when providing care and psychosocial and home management support? How do you think your psychosocial skills influence your professional work? How do situations involving the risk or vulnerability of dependants affect you or have they affected you in your personal life?
Closing	Would you like to add anything else? Thank you for your participation. You will receive the results of the study once it has been completed.

**Table 3 healthcare-11-02678-t003:** Themes, subthemes and units of meaning.

Theme	Subtheme	Units of Meaning
The role of the healthcare support worker in caring for dependent people.	Autonomy as a synonym for success.	Providing care, domestic and psychological support to the patient, ensuring family collaboration, promoting personal autonomy, adapting necessary support, and maintaining and improving quality of life.
	The positive social impact of educational and professional success as a result of a vocational calling.	Striving for your vocation, caregiver assertiveness, effective communication, trust, personal growth, professional efficiency, empathy, active listening, hope, serenity, humanism, personal maturity, and compassionate caregiver.
The training process of the healthcare support worker.	Practice-based learning	Effective dual training, realistic dual training, employment promotion, career development, meaningful learning, effective and essential internships, academic bridge, and complementary theory and internships.
	Training as the foundation of professional success.	Extend personal hygiene care and first aid, multidisciplinary training, need for more practical content, and health terminology.
Vulnerability of the healthcare support worker: risks and needs.	The risk of assault: an unsafe reality.	Physical assaults, patient mistrust, distortion of reality, social stigma, social denial, discerning between professional and personal matters, communication barriers, ideological barriers, physical barriers, and professional undervaluation.
	The risk of emotional overload and the need to learn to manage emotions.	Avoiding emotional dependency, avoiding professional interference, psychological strength, psychological impact, need for human psychology, and reinforcing psychological skills.

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
