# Peer review of "Learning Experiences of Future Healthcare Support Workers Enrolled in a Dual Mode Vocational Training Programme"

_healthcare, 2023, doi:10.3390/healthcare11192678_

Round 1
Reviewer 1 Report
Please find attached file for some improvement suggestions.
It needs proof-reading and some minor spelling check.
The paper also needs rigorous editing.

See above.
Author Response
Thank you very much for your contributions.
- All changes have been made to the text.
- Regarding the question: pre-analytical intuitions
The interviews were conducted by one researcher with an approximate duration of 45 minutes. During both techniques, notes were made with pre-analytical institutions, which means that they were pre-annotations (what was thought to be answered).
- To the question: What are these?
For example: I14, FG6
These are the codings of the participants, I refers to the Interviews and FG to the Focus Group.

Reviewer 2 Report
This study investigated Learning experiences of future healthcare support workers enrolled in a Dual Mode Vocational Training programme.
The study, which findings suggest that educational institutions should focus training programmes on the practice and development of psychosocial skills, is very interesting.
Nevertheless, I have some comments and suggestions for authors, listed below:
Major comments
- Lines 45-54: The authors could do more to explain the classic, standard program, such as the different courses on offer...
- Lines 60-66: « A qualitative study was designed based on Gadamer's hermeneutic philosophy, which highlights the dialogical character of understanding [15,16]. Language allows us to reveal the human essence: through linguistic interpretation we gain access to truth and knowledge of the world [17]. For Gadamer, it is not possible to disregard the researcher's prior conceptions. Therefore, researchers must ensure that the phenomenon they intend to study can be experienced in the lifeworld and they must clarify their prior understanding of the phenomenon [15,18]. »
The authors should move this part to the introduction. Indeed, the concept must be explained before the methods section. Moreover, the authors should explain how the made this choice (for example, they could compare several approaches…).
- Line 82, Table 1: The authors noted the marital status. Could they explain this choice? Why is it important to specify marital status?
- Line 256-257: In the discussion section, the authors say that “A healthcare support worker provides support to people in need as reflected in other studies [22].”
But they present only one study. They should either give several study references, or rephrase the sentence by saying a sentence like:
ð “A healthcare support worker provides support to people in need as reflected in other studies, such as Wu et al. [22].”
Minor comments (comments on the whole manuscript)
Authors’adresses
- Line 8: A space is missing before “Facultad”. Moreover, the address should be written in English, as the other addresses.
Abstract
- Lines 18: “in -depth”: the space after “in” should be suppressed.
ð “in-depth”
Background
- Line 35: “19.77%”: Numbers in beginning of sentence have to be written in letters. The authors could reword the sentence to avoid putting the percentage at the beginning of the sentence.
- Line 48-49: Skipping a line is not shown here. The authors should remove it.
Materials and methods
- Line81: “(Tabla1)”: The authors should write “(Table 1)”.
- Line 83: “Nota:” should be written “Note.” in italics
Results
- Line 126-Table 3: A period is missing at the end of the sentence in the first column, line 4.
- Line 126-Table 3: “Vulnerability of the healthcare support worker: Risks and needs.”. A capital letter is not required after a colon.
Discussion
- Line 261-262: “an increased demand for health care.[24]” . The period should follow the quotation in square brackets.
ð “an increased demand for health ca-re[24].”
Author Response
Thank you very much for your contributions

Reviewer 3 Report
Major points
· The authors have declared that the study followed the COREQ guidelines. However, several items of the COREQ checklist have been missed, including credential, occupation, gender, experience and training of the researchers, relationships with participants, method to approach participants, etc.
· The process of analysis is not clear to me. Lines 106-108 state that first quotations were extracted and then they were assigned codes that were finally grouped into units of meaning, sub-themes and themes. Were only the quotations reported in the paper coded? Authors should better clarify the process of analysis. Readers cannot trust the study findings if data analysis is unclear. Moreover, the authors stated that the study design was based on Gadamer’s hermeneutic philosophy; however, it seems to me that the data analysis is more consistent with a qualitative descriptive approach. I perceived a mismatch between the study design declared and the process of data analysis followed.
Minor points
· In general, there are too many acronyms that make reading difficult. For example, secondary school is abbreviated as SS even if it appears only twice. My suggestion is to spell out how much acronyms as possible to improve readability.
· Line 48. New paragraph not needed.
· Paragraph 2.1. Reference 15 and 16 should not be employed to sustain the statement. In my opinion, only reference 17 is fine.
· Lines 66-68 just repeat the study aim and do not fit with the design section. I suggest removal.
· Inclusion criteria overlap with exclusion criteria. Exclusion criteria should be something more and/or different from inclusion criteria. This point should be revised.
· How many students were approached? 23 students consented out of how many?
· Line 87. Specify the number of students involved in the focus group.
· Line 122. Not scientific statement. Usually, in scientific papers authors should avoid to speak personally.
· The participant reference reported close to each quotation does not match with the anonymised code in table 1 (for example FG6 instead of P6-FG).
· More comment and insight is needed for the second limitation (lines 302-305).
· Lines 306-310 should be moved to the discussion or the conclusions.
Moderate editing of English language required. Some statements are difficult to follow and even if they are grammatically correct, they sound as a translation from the native language to English.
Author Response
Thank you very much for your very pertinent contributions.

Round 2
Reviewer 3 Report
The authors have addressed most but not all of my previous comments. I believe that some of them are important to improve the understandability of the paper, therefore they are again underlined. Specifically:
Lines 57-63, capital letters should be deleted.
Please, add information regarding COREQ items in the text or add the complete COREQ checklist as an appendix.
Line 68, please report only the number referring to the reference, not the overall reference.
Lines 78-84 lack reference.
Exclusion criterium 1 overlaps with inclusion criterium 1.
Line 96. I do not understand this sentence and why it has been place in the results section. It sounds more like a method.
Please, report in the text the number of students approached.
Line 139. Unfinished sentence.
Lines 319-321. I’m unclear with this sentence and do not understand how it supports reasoning. I suggest its removal.
Conclusions. In addition to shaping future research, authors should wrap up the main study findings.
Revision of English has not been satisfactory. The involvement of a native English-speaking editor is required. Please, provide a certificate of English revision.
Author Response
The authors have responded to most of my previous comments, but not all of them. I believe that some of them are important to improve the comprehensibility of the document, so they are underlined again. In particular:
Thank you very much for your contributions.
Lines 57-63, capital letters should be deleted.
The capital letters have been removed.
Please add the information regarding the COREQ items in the text or add the complete COREQ checklist as an appendix.
Information has been added in the manuscript.
Line 68, please report only the reference number, not the overall reference.
It has been changed.
Lines 78-84 have no reference.
The reference has been added.
Exclusion criterion 1 overlaps with inclusion criterion 1.
Exclusion criteria has been removed.
Line 96. I do not understand this sentence and why it has been placed in the results section. It seems more like a method.
It has been removed.
Please report in the text the number of students addressed.
Has been added in the text.
Line 139. Unfinished sentence.
Sentence has been completed.
Lines 319-321. This sentence is not clear to me and I do not understand how it supports the reasoning. I suggest its deletion.
Deleted.
Conclusions. In addition to shaping future research, the authors should summarise the main conclusions of the study.
The conclusions section contains the main findings of the study.